# Effects of the pan-caspase inhibitor Q-VD-OPh on human neutrophil lifespan and function

Lisa Khuu[1], Alisha Pillay[1], Allan Prichard[1,2¤], Lee-Ann H. Allen[1,2]*

1 Department of Molecular Microbiology and Immunology, University of Missouri, Columbia, Missouri, United States of America, 2 Department of Microbiology and Immunology, University of Iowa, Iowa City, Iowa, United States of America

¤ Current address: AstraZeneca, Gaithersburg, Maryland, United States of America
* lee-ann.allen@health.missouri.edu

**Data Availability Statement:** All relevant data are contained within the manuscript and its Supporting information files.

**Funding:** This work was supported, in part, by funds from the US government, including National

## Abstract

Human neutrophils are abundant, short-lived leukocytes that turn over at a rate of approximately $10^{11}$ cells/day via a constitutive apoptosis program. Certain growth factors, inflammatory mediators and infectious agents can delay apoptosis or induce neutrophils to die by other mechanisms. Nonetheless, a large body of data demonstrates that apoptosis of untreated neutrophils typically ensues within 24 hours of cell isolation and *in vitro* culture. At the molecular level apoptosis is driven by executioner caspase-3, and during this process cell proinflammatory capacity and host defense functions are downregulated. We undertook the current study to determine the extent to which human neutrophil viability and function could be prolonged by treatment with the non-toxic, irreversible, pan-caspase inhibitor Q-VD-OPh. Our data demonstrate that a single 10 µM dose of this drug was sufficient to markedly prolong cell lifespan. Specifically, we show that apoptosis was prevented for at least 5 days as indicated by analysis of nuclear morphology, DNA fragmentation, and phosphatidylserine externalization together with measurements of procaspase-3 processing and caspase activity. Conversely, mitochondrial depolarization declined despite abundant Myeloid Cell Leukemia 1 (MCL-1). At the same time, glutathione levels were maintained and Q-VD-OPh prevented age-associated increases mitochondrial oxidative stress. Regarding functional capacity, we show that phagocytosis, NADPH oxidase activity, chemotaxis, and degranulation were maintained following Q-VD-OPh treatment, albeit to somewhat different extents. Thus, a single 10 µM dose of Q-VD-OPh can sustain human neutrophil viability and function for at least 5 days.

## Introduction

Neutrophils (polymorphonuclear leukocytes, PMNs) are essential for innate host defenses and are the most abundant leukocyte type in the bloodstream of humans [1, 2]. These cells respond rapidly to infection executing effector functions that include chemotaxis, phagocytosis and killing of invading microbes via degranulation and production of toxic reactive oxygen species (ROS) [2]. Produced at a rate of approximately $10^{11}$ cells per day in humans, neutrophils are

Institutes of Health R01 AI119965 awarded to L-AA. Part of this study was also supported by funds awarded to L-AA by the University of Missouri School of Medicine. The funders had no role in study design, data collection and analysis, decision to publish, or preparation of the manuscript.

**Competing interests:** The authors have declared that no competing interests exist.

inherently programmed to undergo a tightly regulated and immunologically silent constitutive apoptosis program [1–5]. Inasmuch as the precise lifespan of these cells in circulation remains a matter of debate, there is extensive evidence that apoptosis ensues within 24 hours after isolation and *in vitro* culture [6–8]. Hallmarks of neutrophil apoptosis include disappearance of the prosurvival regulatory factor Myeloid Cell Leukemia 1 (MCL-1), translocation of the proapoptotic factor B-Cell Lymphoma 2 (BCL-2)-Associated X (BAX) to execute mitochondrial outer membrane permeabilization (MOMP), activation of caspases-8, -9, -7 and -3, phosphatidylserine (PS) accumulation in the outer leaflet of the plasma membrane, DNA fragmentation and nuclear condensation [1, 2]. In parallel, neutrophil proinflammatory capacity and host defense functions are downregulated and release of cytoplasmic enzymes and DAMPs (damage-associated molecular patterns) is prevented [3]. Under certain conditions and disease states neutrophils can be reprogrammed to undergo other forms of cell death such as NETosis, a lytic form of death that is defined by release of decondensed DNA coated with granule proteins as a neutrophil extracellular trap (NET), necroptosis or pyroptosis, the latter of which is mediated by caspase-1, alone or together with caspases-4 and -5 in humans or caspase-11 in mice [3, 9, 10]. Additionally, PANoptosis, a hybrid form of cell death that utilizes subsets of molecules from pyroptosis, apoptosis and necroptosis pathways, has also been described [11].

The extent to which neutrophil lifespan can be prolonged is incompletely defined. Growth factors such as Granulocyte Colony Stimulating Factor (G-CSF), Granulocyte-Macrophage Colony Stimulating Factor (GM-CSF) and certain inflammatory mediators delay neutrophil apoptosis by approximately 12 hours, thereby helping to ensure that viable cells reach sites of infection for optimal microbe killing [12, 13]. On the other hand, intracellular pathogens such as *Anaplasma phagocytophilum*, *Chlamydia pneumoniae*, *Francisella tularensis* and *Leishmania* parasites significantly delay apoptosis onset to maximize their own replication [1]. Q-VD-OPh (hereafter referred to as QVD) is a cell-permeable, irreversible, and non-toxic true pan-caspase inhibitor that prevents apoptosis of many cell types *in vivo* and *in vitro* [14–17]. Previous studies provide independent evidence to demonstrate that QVD prevents apoptosis of human neutrophils during the first 20–48 h after cell isolation, the longest time point examined [16, 18, 19].

We undertook the current study to determine the lifespan and functional capacity of human neutrophils treated with a single 10 μM dose of QVD immediately after isolation. Our data demonstrate that neutrophils treated in this manner survived for at least 5 days *in vitro* and retained their capacity for phagocytosis, chemotaxis, degranulation and ROS production despite mitochondrial depolarization.

## Materials and methods

### Ethics statement

Healthy adults were recruited as volunteer blood donors for this study. All blood donors provided written informed consent prior to enrollment in conjunction with protocol #2031144 that was originally approved by the University of Missouri Institutional Review Board on September 30, 2020, and has been renewed annually. Recruitment of blood donors for our studies opened on October 21, 2020 and is on-going. Our donor pool includes similar numbers of men and women and reflects the composition of the university community.

### Isolation of human neutrophils

Heparin-anticoagulated whole blood obtained by venipuncture was fractionated as previously described [20]. In brief, leukocytes were enriched from whole blood using dextran to sediment the erythrocytes. Next, neutrophils were separated from peripheral blood mononuclear cells

(monocytes and lymphocytes) by Ficoll-Hypaque (Cytiva, Marlborough MA) density gradient separation. Finally, residual erythrocytes in the isolated neutrophil fraction were removed by hypotonic lysis. Cell preparations were routinely >95% neutrophils with eosinophils as the major contaminant.

## Neutrophil culture and drug treatments

Neutrophils were resuspended in 2-[4-(2-hydroxyethyl)piperazin-1-yl]ethanesulfonic acid (HEPES)-buffered Roswell Park Memorial Institute (RPMI)-1640 medium with or without phenol red that was supplemented with 2 mM L-glutamine (all from Lonza, Walkersville MD, distributed by Fisher Scientific, Pittsburgh PA) and 10% heat-inactivated fetal bovine serum (FBS) (Gibco, Grand Island, NY) at $1x10^6$ cells/ml in polypropylene tubes unless otherwise stated. Neutrophils were incubated at 37˚C under humidified conditions and 5% $CO_2$ until the indicated time points. As indicated, 10 µM Q-VD-OPh (R&D Systems, Minneapolis MN) was added at the start of the incubation period. In selected experiments lower doses of QVD (0.1 µM and 1 µM) were also tested. Preliminary experiments demonstrated lack of effect of the DMSO diluent on assays of cell function and apoptosis (not illustrated). Where applicable, neutrophils were treated with 10 µg/ml cycloheximide, 5 µM MG-132 (both from MedChem-Express, Monmouth Junction NJ), or 5 µM Bortezomib (MilliporeSigma, St. Louis MO) after 24 h of aging or QVD treatment. In all cases, replicate experiments utilized neutrophils from different donors.

## Kinetics of nuclear condensation

Aliquots of neutrophil suspensions were cytocentrifuged onto 12mm round glass coverslips in triplicate, stained using PROTOCOL™ Hema-3 reagents (Fisher Scientific), and mounted onto glass slides with Permount (Fisher Scientific) as we previously described [8]. Images of stained cells were acquired using a Leica DMi8 light microscope fitted with a 63x-oil objective (Leica Microsystems, Buffalo Grove IL). At least 100 cells per coverslip from 10 random fields of view were examined per experiment and condition. Nuclear morphology was scored as normal/lobed or condensed/apoptotic as we described [8, 21].

## Quantitation of DNA fragmentation

DNA fragmentation was measured using APO-BRDU™ (BD Pharmingen, Franklin Lakes NJ), a modified TUNEL (terminal deoxynucleotidyl transferase dUTP nick-end labeling) assay, according to the manufacturer's guidelines and as previously described [8]. Briefly, cells were fixed with 1% paraformaldehyde, pelleted, washed with phosphate-buffered saline (PBS), and then incubated at the specified concentration in 70% ethanol for at least 18 h. After washing to remove ethanol, cells were incubated in DNA labeling solution at 37˚C for 1 h, washed, resuspended in staining buffer and then incubated for 30 min with fluorescein isothiocyanate (FITC)-conjugated anti-BrdU (bromodeoxuridine) antibodies in the dark at room temperature. Stained cells were analyzed immediately by flow cytometry. For each sample, at least 10,000 events were collected on a Beckman Coulter CytoFLEX flow cytometer (Indianapolis, IN) and data were analyzed using FlowJo v.10 software.

## Flow cytometry

**Annexin V-FITC staining.** Phosphatidylserine (PS) externalization and loss of plasma membrane integrity were quantified using Annexin V-FITC/propidium iodide (PI) dual staining as we described [8]. In brief, cell pellets were co-stained with Annexin V-FITC and PI

(both from BioVision, Waltham, MA) in Annexin V binding buffer (10 mM HEPES, 140 mM NaCl, 2.5 mM CaCl$_2$ pH 7.4) for 5 min in the dark. Thereafter, samples were analyzed by flow cytometry. In each case, at least 10,000 events were collected per sample on a Beckman Coulter CytoFLEX. Standard compensation was performed using single-stained 24 hour-aged PMNs and data were analyzed using FlowJo v.10 software.

**Mammalian target of rapamycin (mTOR) and phospho-mTOR.** At the indicated time-points, PMNs were fixed in 4% paraformaldehyde (PFA) for 15 min in the dark and then permeabilized with 90% methanol in PBS for 10 min in the dark, on ice. Thereafter, PMNs were stained with anti-mTOR and Ser2448 phospho-mTOR antibodies (Cell Signaling 7C10 #2983 and D9C4 #5536, respectively), conjugated to Zenon™ AF488 Rabbit IgG (Thermo Fisher) according to manufacturer's guidelines. PMNs were quenched with a 1:1 mixture of FBS and PBS, pelleted, resuspended in PBS, and analyzed on a Beckman CytoFLEX as described above.

## Immunoblotting

PMNs were incubated at 2x10$^6$ cells/mL, and 10 million cells were used per sample and condition. Cell suspensions were pelleted by centrifugation at 300xg for 5 min at room temperature, resuspended in 1 mL of Hanks' Balanced Salt Solution without divalent cations (HBSS-/-, Corning, Manassas, VA), and then treated with diisopropylfluorophosphate (DFP; Millipore-Sigma) for 20 min at room temperature. Cells were lysed by incubation on ice for 10 min in radioimmunoprecipitation assay (RIPA) buffer (Thermo Fisher, Waltham, MA) supplemented with protease and phosphatase inhibitor cocktails (MilliporeSigma and Thermo Fisher, respectively) and additional aprotinin, levamisole, AEBSF (4-(2-aminoethyl) benzene sylfonyl fluoride hydrochloride), pepstatin A (all from MilliporeSigma) and leupeptin (Thermo Fisher) [18, 22, 23] and then clarified by centrifugation. Proteins in each clarified lysate were quantified using Pierce BCA (bicinchoninic acid) Protein Assay Kits (Thermo Fisher) according to the manufacturer's protocols with the absorbance at 562 nm measured on a CLARIOstar Plus (BMG LabTech, Cary, NC).

After boiling in sodium dodecyl sulfate-polyacrylamide gel electrophoresis (SDS-PAGE) sample buffer with reducing reagent, 20 μg aliquots of protein lysates were loaded into Bis-Tris gels in 2-(N-morpholino)ethanesulfonic acid (MES) SDS running buffer supplemented with antioxidant (all from Thermo Fisher), alongside Precision Plus Protein Standards (Bio-Rad, Richmond CA). After electrophoresis, proteins were transferred onto polyvinylidine fluoride (PVDF) membranes using the Trans-blot Turbo Transfer System (Bio-Rad) and incubated for 1 h at room temperature in blocking buffer [Tris-buffered saline (TBS, pH 7.4) supplemented with 0.1% Tween-20 and 5% bovine serum albumin (BSA) or 5% nonfat milk] and then probed overnight at 4°C with mouse anti-human caspase-3 (BioVision #3004, Waltham MA), rabbit anti-XIAP (X-linked inhibitor of apoptosis protein) (#14334, Cell Signaling, Danvers MA), rabbit anti-MCL-1 (Proteintech #16225-1-AP, Rosemont IL) or rabbit anti-LC3B (Abgent #AP1806a, San Diego, CA) antibodies, each diluted 1:1,000 in blocking buffer. Secondary horseradish peroxidase-conjugated antibodies (Cell Signaling #7074S and #7076S) were diluted 1:2,000 in blocking buffer and incubated with PVDF membranes for 1 hour at room temperature. Bands were detected using West Pico chemiluminescent substrate (Thermo Fisher) and imaged on a LI-COR Odyssey XF (LI-COR Biosciences, Lincoln NE). Blots were stripped and re-probed with anti-GAPDH (MilliporeSigma #CB1001) as a loading control. Bands were quantified by densitometry using ImageJ. Uncropped images of all immunoblots are provided in S1 Raw images.

## Caspase activity assays

Activities of caspases 1, 3, 5, and 6 were measured using the Caspase-Glo$^®$ 1 Inflammasome Assay Kits (Promega, Madison WI) according to the manufacturer's directions. Briefly, cells were cultured in HEPES-buffered phenol red-free RPMI-1640 medium and then diluted to $5 \times 10^5$ cells/mL. Equal volumes of cell cultures and reagent containing the luminogenic caspase substrate Z-WEHD-aminoluciferin and MG-132 proteasome inhibitor were loaded into a white 96-well plate in triplicate and incubated at room temperature with shaking for 1 h in the dark. Luminescence was quantified using a CLARIOstar Plus with top reads at 0 and 30 min and a media only blank.

## Analysis of mitochondrial integrity, abundance and superoxide

Dissipation of mitochondrial membrane potential in live cells was quantified using MitoProbe™ JC-1 Assay Kits (Thermo Fisher) as previously described and in accordance with the manufacturer's instructions [21, 24]. Total mitochondrial mass was measured using MitoTracker™ Deep Red FM and superoxide in mitochondria of live cells was quantified using MitoSOX™ Red Mitochondrial Superoxide Indicator (both from Thermo Fisher). In each case, cells were stained according to the manufacturer's directions and then analyzed by flow cytometry using a Beckman Coulter CytoFLEX as described above.

## Quantitation of nicotinamide adenine dinucleotide (NAD) and glutathione

Total cellular NAD levels were measured using NAD/NADH Cell-Based Assay Kits (Cayman Chemical, Ann Arbor MI) according to manufacturer's guidelines with minor modifications. Cells were permeabilized as instructed and centrifuged at 10,000g for 10 min. Clarified supernatants were stored at -20°C until use, at which point thawed samples and standards were aliquoted into 96-well microtiter plates in triplicate. Reaction solution was added and mixed for 90 min at room temperature with gentle rocking while protected from light. Absorbances at 450 nm were detected on a CLARIOstar Plus.

Ratios of glutathione (GSH) to glutathione disulfide (GSSG) were measured using GSH/GSSG-Glo™ Assay kits (Promega) according to manufacturer's guidelines. In short, at the selected timepoints cells were concentrated to $5 \times 10^7$ cells/mL in PBS and aliquoted in duplicate into 96-well white-walled microtiter plates with no-cell controls as the blank. Kit reagents were added as instructed with gentle mixing and incubated at room temperature covered with foil. Luminescence was detected on a CLARIOstar Plus and ratios were calculated as instructed.

## Confocal microscopy

Neutrophils were attached to acid-washed, serum-coated glass coverslips and then fixed and processed for confocal microscopy using our established methods [25] with minor modifications. Specifically, cells were fixed in 4% PFA and then permeabilized with a 1:1 methanol:ethanol. After incubation in blocking buffer (PBS supplemented with 0.5 mg/mL NaN$_3$, 5 mg/ml bovine serum albumin (BSA) and 10% horse serum), cells were stained to detect BAX using anti-BAX (BD Pharmingen #556467) and MnSOD (manganese superoxide dismutase) using anti-MnSOD (Thermo Fisher #PA1-125) or p62 using rabbit anti-SQSTM1/p62 antibodies (Cell Signaling #7695, D10E10) Primary antibodies and Alexa Fluor (AF)488- and AF647-conjugated F(ab')$_2$ secondary antibodies (Jackson ImmunoResearch, West Grove PA) diluted 1:200 in blocking buffer. Each incubation was 1 h at room temperature and coverslips were washed after each antibody incubation in PBS supplemented with 0.5 mg/ml NaN$_3$ and 5 mg/ml BSA. Coverslips were attached to slides using ProLong Diamond anti-fade mounting

medium with 4',6-diamidino-2-phenylindole (DAPI) (Thermo Fisher). Images were acquired using a Leica Stellaris 5 confocal microscope with a 63x-oil objective and LAS-X software (Leica Microsystems).

## Quantitation of phagocytosis

Yeast zymosan particles in HBSS-/- were dispersed by water bath sonication, mixed with an equal volume of pooled human serum, and then opsonized by incubation for 30 min at 37˚C. After washing, opsonized zymosan (OpZ) particles were resuspended in complete tissue culture medium and fed to PMNs at a ratio of 4 particles/cell for 15 min at 37˚C with nutation. At the end of the incubation period, cells were cytocentrifuged onto coverslips and stained with Hema-3 reagents as described above. The percentage of neutrophils that ingested OpZ and the number of OpZ particles per phagocytic cell were quantified using light microscopy as we previously described [23].

## EZ-TAXIScan™ live cell imaging

PMNs were pelleted and resuspended at $10^6$/mL in EZT buffer [HBSS with calcium and magnesium (HBSS+/+) supplemented with 20 mM HEPES and 0.1% human serum albumin (HSA)] and EZ-TAXIScan™ movies were obtained as we previously described [23, 26]. In brief, chips containing 6-channel migration chambers were assembled and top chambers were loaded with cell suspensions. Chemokine gradients were formed by injecting 1 μM fMLF (N-formyl-L-methionyl-L-leucyl-phenylalanine) (Santa Cruz, Dallas TX) into the bottom chamber of each channel. Movies were recorded with 1 sec time lapse intervals between channels for 60 min (3 images/channel/min). Movies were processed in ImageJ to a set time of 20 sec. Migration speeds (instantaneous velocities) and chemotactic indices (directionality of migration) of individual cells were calculated using established procedures [23, 26]. In each case, 20 individual cells per experiment and condition were manually tracked. Instantaneous velocities indicate migration speeds whereas the ratio of net path length toward the fMLF attractant relative to the total path length is the chemotactic index (CI). Perfect forward linear migration has a CI of 100, increased random migration reduces the score and migration away from the attractant earns a negative score.

## Respiratory burst

The kinetics of neutrophil NADPH oxidase-derived ROS production was measured using the luminol-enhanced chemiluminescence assay as we previously described with minor modifications [27]. In brief, PMNs were pelleted and resuspended at $1x10^7$/mL in phenol red-free HEPES-buffered RPMI-1640 medium. Serum-coated white 96-well plates were loaded in triplicate for each sample with each well containing $1x10^6$ cells, 50 μM luminol (MilliporeSigma) and 4% HSA [28]. Plates were heated to 37˚C in a CLARIOstar Plus and luminescence was read from the top of each well at 30 sec intervals for 1 h. NADPH oxidase activity was stimulated by the addition of phorbol myristate acetate (PMA, Sigma-Aldrich) to 200 nM final concentration after a 15 min luminol-loading period.

## Degranulation

For these experiments, neutrophils isolated using our standard procedures were ultra-purified (>99% pure) using EasySep™ Human Neutrophil Isolation Kits (Promega) which deplete contaminating leukocytes using antibody-coated beads according to the manufacturer's instructions. At the indicated timepoints, cell culture supernatants were collected, passed through

0.22 μm filters to remove any intact cells, and stored at -20˚C. Supernatant were gently thawed, and elastase was quantified using Human PMN Elastase DuoSet ELISA kits (R&D Systems) according to the manufacturer's. Results were detected on a CLARIOstar Plus (BMG Labtech) with top reads of absorbance at 450 nm.

### Analysis of neutrophil surface markers

PMN surface markers were quantified by flow cytometry after staining with antibodies in a DURAClone IM Granulocyte kit (Beckman Coulter) according to the manufacturer's instructions.

### Statistical analysis

All data were analyzed using GraphPad Prism version 9.3 software. Data are shown as the mean and standard deviation (SD) of three or more independent determinations. Data were analyzed using two-way ANOVA with Tukey's multiple comparisons post-test or Student's t-test as indicated in each figure legend. In all cases, $p$ values less than 0.05 were considered statistically significant. Nonsignificant differences are not labeled.

## Results

### Q-VD-OPh ablates neutrophil apoptosis for at least five days

Neutrophil apoptosis has been extensively studied and constitutive apoptosis of aged cells is driven by the intrinsic apoptosis pathway [1]. MOMP is a key initiating event, as release of cytochrome *c* from the intermembrane space into the cytosol triggers assembly of the apoptosome, a platform for activation of procaspase-9 [1, 2]. Active caspase-9 then cleaves and activates procaspase-3 to execute a cell death program notable for accumulation of PS in the external leaflet of the plasma membrane, DNA fragmentation and nuclear condensation [2]. Inasmuch as PS externalization is considered a hallmark of early apoptosis, loss of plasma membrane integrity accompanies progression to late apoptosis/secondary necrosis. Herein, we used a series of established assays to quantify the ability of a single 10 μM dose of QVD to prevent human neutrophil progression to apoptosis over 5 days in suspension culture.

As a first approach, we cytocentrifuged neutrophils onto coverslips, stained the cells with Hema-3 reagents, and used light microscopy to assess acquisition of a condensed/apoptotic nuclear morphology. Representative images are shown in Fig 1A and pooled data from three independent experiments performed in triplicate are shown in Fig 1B. As expected, more than 98% of all neutrophils exhibited healthy, lobed nuclei shortly after isolation (PMN day 0, P0) and aged PMNs progressed to apoptosis over 24 h (P1) to 48 h (P2) in agreement with published data [8]. By day 3 (P3), total cell numbers were diminished, but 92% of the cells that remained contained condensed nuclei. As few aged cells remained on day 3, this time point was not routinely included in subsequent assays. By contrast, QVD nearly ablated this change in nuclear morphology, as ≤10% of drug-treated cells had condensed nuclei at all timepoints examined over 5 days (Q1-Q5, Fig 1A and 1B).

To assess the upper limit of QVD efficacy, we incubated some cells in drug-containing medium for 6–10 days. Over this time course, neutrophils became increasingly fragile and accumulated cytoplasmic vacuoles. Nonetheless, nuclear condensation was not observed, and cells retained a normal, lobed morphology or appeared somewhat hypersegmented (5 or more nuclear lobes per cell) (S1 Fig). Because of their fragility, and because cell vacuolation may indicate diminished fitness or nutrient scavenging via autophagy [1], all subsequent experiments were terminated on QVD day 5.

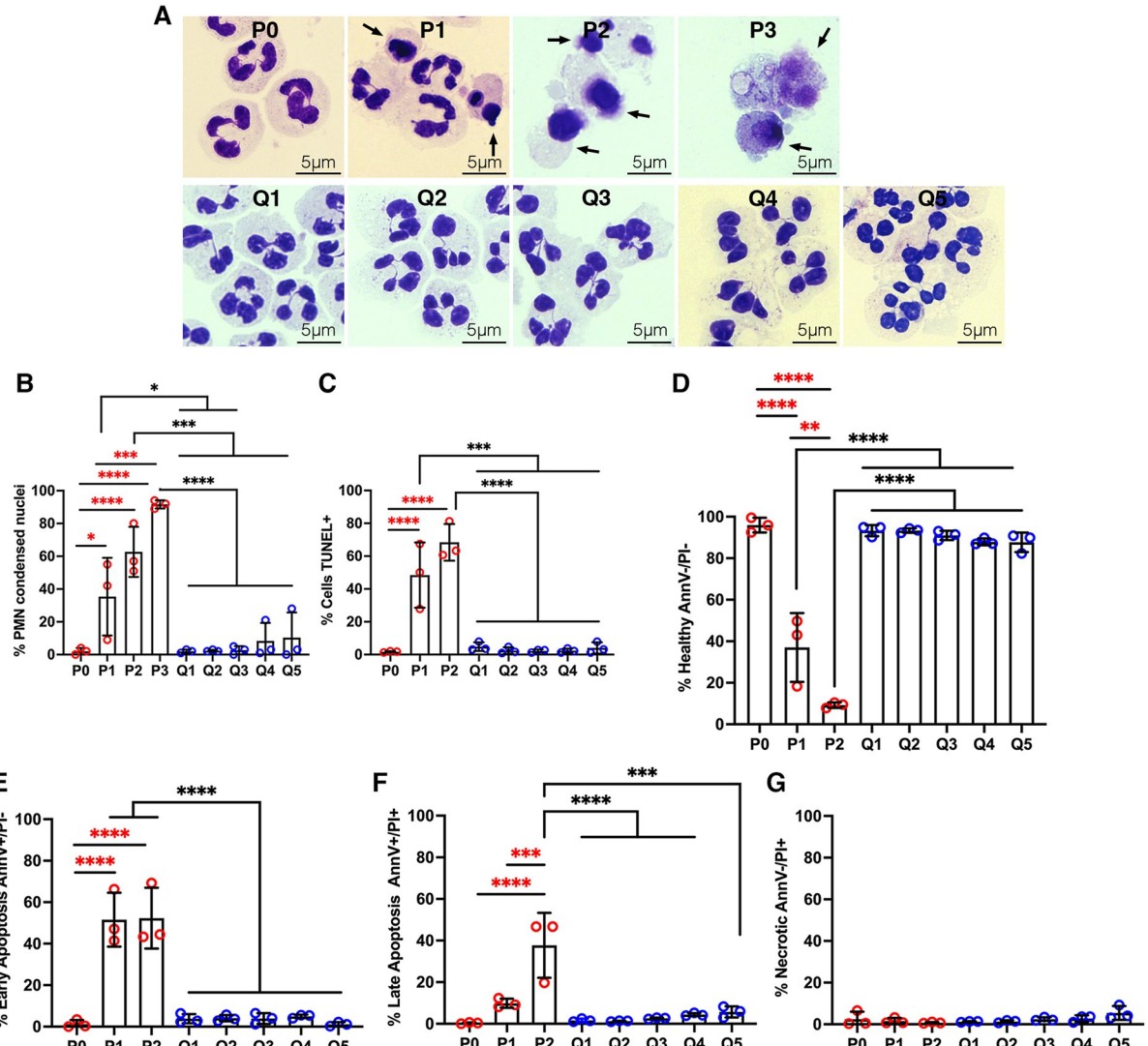

**Fig 1. QVD treatment prevents apoptosis for at least 5 days.** Freshly isolated neutrophils (P0), cells aged for 24 or 48 h (P1, P2) and cells treated with QVD for 1–5 days (Q1-Q5) were assayed for hallmarks of apoptosis. **(A-B)** Nuclear morphology was assayed using Hema-3 staining and light microscopy. Representative images are shown in (A). *Arrows* indicate cells with condensed, apoptotic nuclei. Pooled data in (B) show the percentage of neutrophils with condensed/apoptotic nuclei at each time point and are the mean ± SD of three independent experiments performed in triplicate. **(C)** DNA fragmentation quantified by TUNEL staining at each time point. Data are the mean ± SD of three independent experiments. **(D-G)** Neutrophils were stained with Annexin V-FITC/PI and analyzed by flow cytometry. Graph shows the percentage of healthy cells (Annexin V-/PI-) (D), early apoptotic cells (Annexin V+/PI-) (E), late apoptotic cells (Annexin V+/PI+) (F), and necrotic cells (AnnexinV-/PI+) (G) at each time point and are the the mean ± SD of three independent experiments. All data were analyzed by two-way ANOVA with Tukey's multiple comparisons post-test. *$p<0.05$, **$p<0.01$, ***$p<0.001$, and ****$p<0.0001$.

Next, we detected apoptotic DNA fragmentation using TUNEL staining and flow cytometry [8, 17]. The data in Fig 1C show that fewer than 2% of freshly isolated cells were TUNEL-positive, and DNA fragmentation progressed to 48.4 ± 19.9% positive by 24 h (P1) and 68.4 ± 11.2% positive by 48 h (P2) in aged PMNs. By contrast, DNA fragmentation was nearly ablated (<5% TUNEL-positive) in the presence of QVD.

Annexin V-FITC/PI staining and flow cytometry allows quantitation of four subpopulations of cells within a population: healthy (Annexin V-/PI-), early apoptotic (Annexin V+/PI-),

late apoptotic (Annexin V+/PI+), and necrotic (Annexin-V-/PI+) [8, 21]. By this assay, >95% of freshly isolated PMNs (P0) were healthy, and this declined to 37.0 ± 16.5% by 24 h (P1) and 9.2 ± 1.3% by 48 h (P2) (Fig 1D) in parallel with progression of aged cells to early and late apoptosis over 24–48 h (P1 and P2) (Fig 1E and 1F). Thus, 63.0 ± 16.5% of aged PMNs were in early or late apoptosis at 24 h, and this increased to 90.8 ± 1.3% by 48 h. By contrast, ≤12% of QVD-treated cells progressed to apoptosis over the 5-day time course examined (Q1-Q5) and necrosis was not observed (Fig 1E–1G). Furthermore, results of additional experiments indicate that the ability to QVD to inhibit apoptosis was dose-dependent, as fewer cells remained healthy on Days 1 and 3 after treatment with 1or 0.1 μM QVD; and the inhibition that was observed waned by Day 5 as demonstrated by Annexin V-FITC/PI staining and analysis of nuclear morphology (S2 Fig). Thus, 10 μM QVD was used in all subsequent experiments.

Processing of procaspase-3 to its mature, active form is essential for apoptosis [29]. Data obtained using an established western blotting assay confirm that procaspase-3 was abundant in freshly isolated PMNs whereas the mature active enzyme was absent or below the limit of detection (Fig 2A and 2B) [16, 30]. As expected, mature caspase-3 was present in aged PMNs lysates by 24 h (P1) and was significantly more abundant by 48 h (P2). Distinctly different results were obtained for cells treated with QVD as, in addition to an absence of mature caspase-3 that resembled freshly isolated cells, the proenzyme declined and disappeared (Fig 2A and 2B). We hypothesized that changes in the dynamics of procaspase-3 synthesis or turnover could account for these results. To test this hypothesis, we performed additional experiments adding the protein synthesis inhibitor cycloheximide (CHX) or the proteasome inhibitors MG-132 and bortezomib to cultures of control and QVD-treated neutrophils at 24 h (P1, Q1) and then collected cell lysates 24 h later (P2, Q2). The data demonstrate that CHX caused a significant decline in procaspase-3 abundance in both aged and QVD-treated neutrophils indicating that this protein is continuously synthesized, whereas the extent of enzyme processing to its mature form appeared unchanged (Fig 2C and 2D). Conversely, proteasome inhibitors had no effect on pro- or mature caspase-3 in aged PMNs yet caused a band of intermediate size (~20kDa) to accumulate selectively in QVD-treated cells (Fig 2C and 2D). These data suggest that the decline of procaspase-3 that ensues in cells treated with QVD for prolonged periods of time can be attributed, at least in part, to proteasomal degradation.

Next, we used a Caspase-Glo® luminescence assay to quantify the activity of caspase-3 as well as caspases-1, 5 and 6. The data obtained demonstrate that caspase activity progressively and significantly increased in aged PMNs over 24–48 h (P1-P2) but remained at or below the baseline seen in freshly isolated PMNs after exposure to QVD (Q1-Q5) (Fig 2E). Taken together, our data demonstrate that QVD prevented caspase activation and apoptosis for at least 5 days.

XIAP is a potent endogenous inhibitor of apoptosis protein that binds directly to caspases-9 and -3 to inhibit their activity in healthy neutrophils [1, 31]. During apoptosis, XIAP is degraded as indicated by the presence of a p30 XIAP fragment in cell lysates [32]. Herein, we confirmed the presence of full-length XIAP in freshly isolated neutrophils and the appearance of a 30 kDa fragment in cells aged for 24–48 h (P1-P2) (S3 Fig), in agreement with published data and the kinetics of caspase activation and processing shown in Fig 2. Our data also show that full-length XIAP was present in QVD-treated cells whereas the p30 fragment was not detected at any examined time point.

**QVD does not prevent mitochondrial depolarization despite abundant MCL-1.** MCL-1 is the major pro-survival BCL-2 family protein in neutrophils and plays a critical role in sustaining mitochondrial integrity [1, 12]. As MCL-1 is short-lived, its continued synthesis is required for PMN survival [1, 2]. Thus, a decline of MCL-1coincides with BAX translocation and subsequent MOMP [2, 33]. We confirmed the presence of MCL-1 in freshly isolated

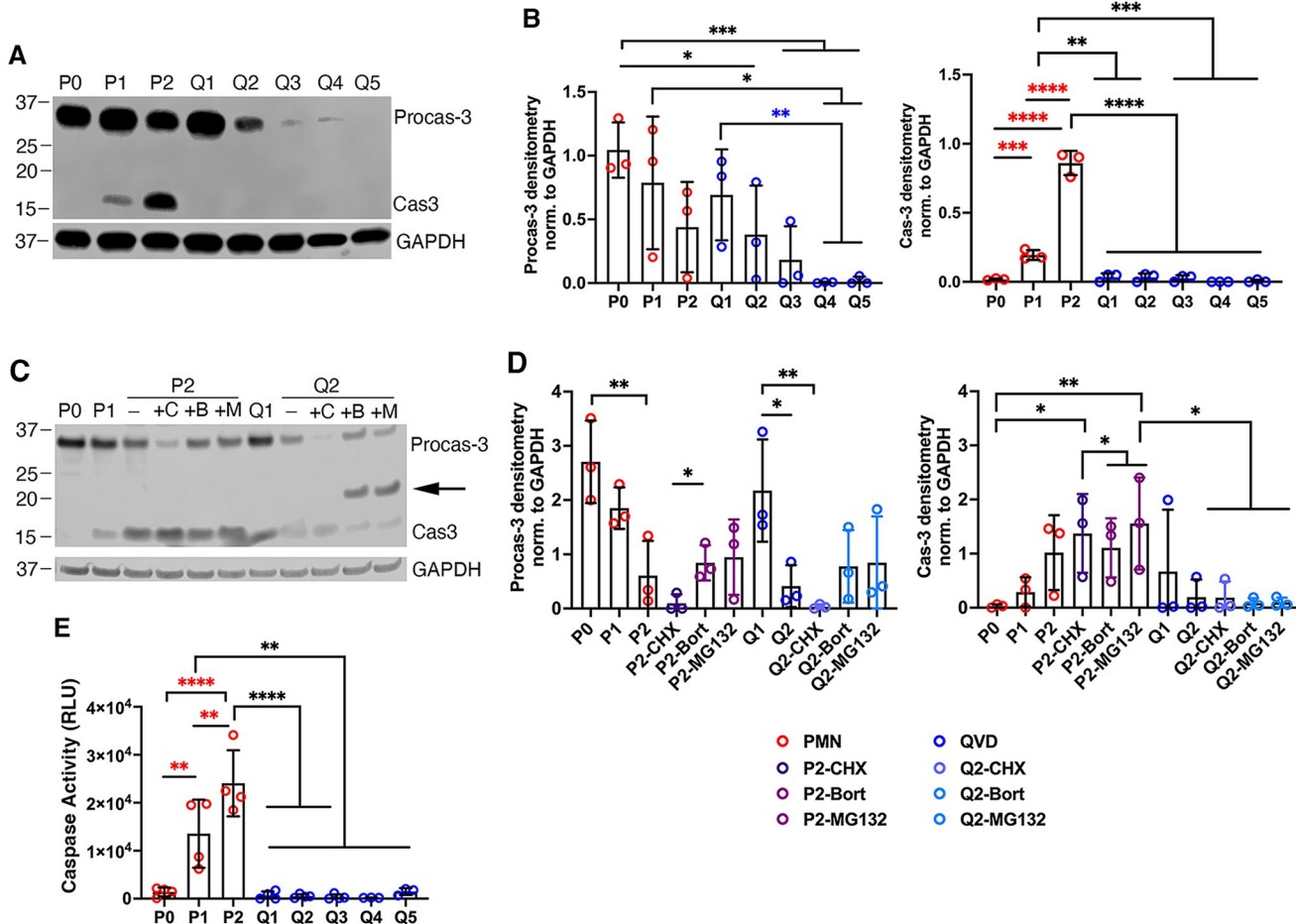

**Fig 2. QVD inhibits caspase processing and activity and leads to procaspase-3 degradation. (A-B)** Pro- and mature caspase-3 were detected in lysates of fresh (P0), 1–2 day aged (P1, P2) and 1–5 day QVD-treated neutrophils (Q1-Q5) by western blotting with GAPDH as the loading control. Representative blots (A) and densitometric quantitation of procaspase-3 and caspase-3 normalized to GAPDH are shown as the mean ± SD of three independent experiments (B). **(C-D)** Representative caspase-3 immunoblot of fresh (P0), 1–2 day aged (P1-P2) and 1–2 day QVD-treated PMNs (Q1-Q2) that were incubated in the absence or presence of 10 μg/ml cycloheximide (C, CHX), 5 μM bortezomib (B, Bort) or 5 μM MG-132 (M) as indicated, with GAPDH as a loading control (C). Normalized densitometric quantitation is the mean ± SD of three independent experiments (D). **(E)** Combined activity of caspases 1, 3, 5, and 6 in fresh (P0), 1–2 day aged (P1-P2) or 1–5 day QVD-treated (Q1-Q5) PMNs quantified using Caspase-Glo® with luminescence shown in relative light units (RLU). For all graphs, data were analyzed by two-way ANOVA with Tukey's multiple comparisons post-test.: *$p<0.05$, **$p<0.01$, and ***$p<0.001$, ****$p<0.0001$.

PMNs and its disappearance from cells that were aged in culture (Fig 3A and 3B), and this correlated with mitochondrial depolarization detected using JC-1 staining and flow cytometry (Fig 3C and S3 Fig). However, distinctly different data were obtained for neutrophils treated with QVD. Specifically, we show that MCL-1 was maintained (Fig 3A and 3B), yet this did not prevent BAX translocation or mitochondrial depolarization (Fig 3C and 3D and S4 Fig). In other cell types, damaged mitochondria can be repaired, cleared by mitophagy, or replaced by the growth and division of healthy organelles [3, 34, 35]. With this in mind, we used Mito-Tracker™ Deep Red together with and flow cytometry to quantify total mitochondrial mass. The data in Fig 3E indicate that mitochondrial abundance was not significantly altered under the conditions of this study. Thus, prolonged QVD treatment is associated with diminished mitochondrial membrane potential but no change in organelle mass.

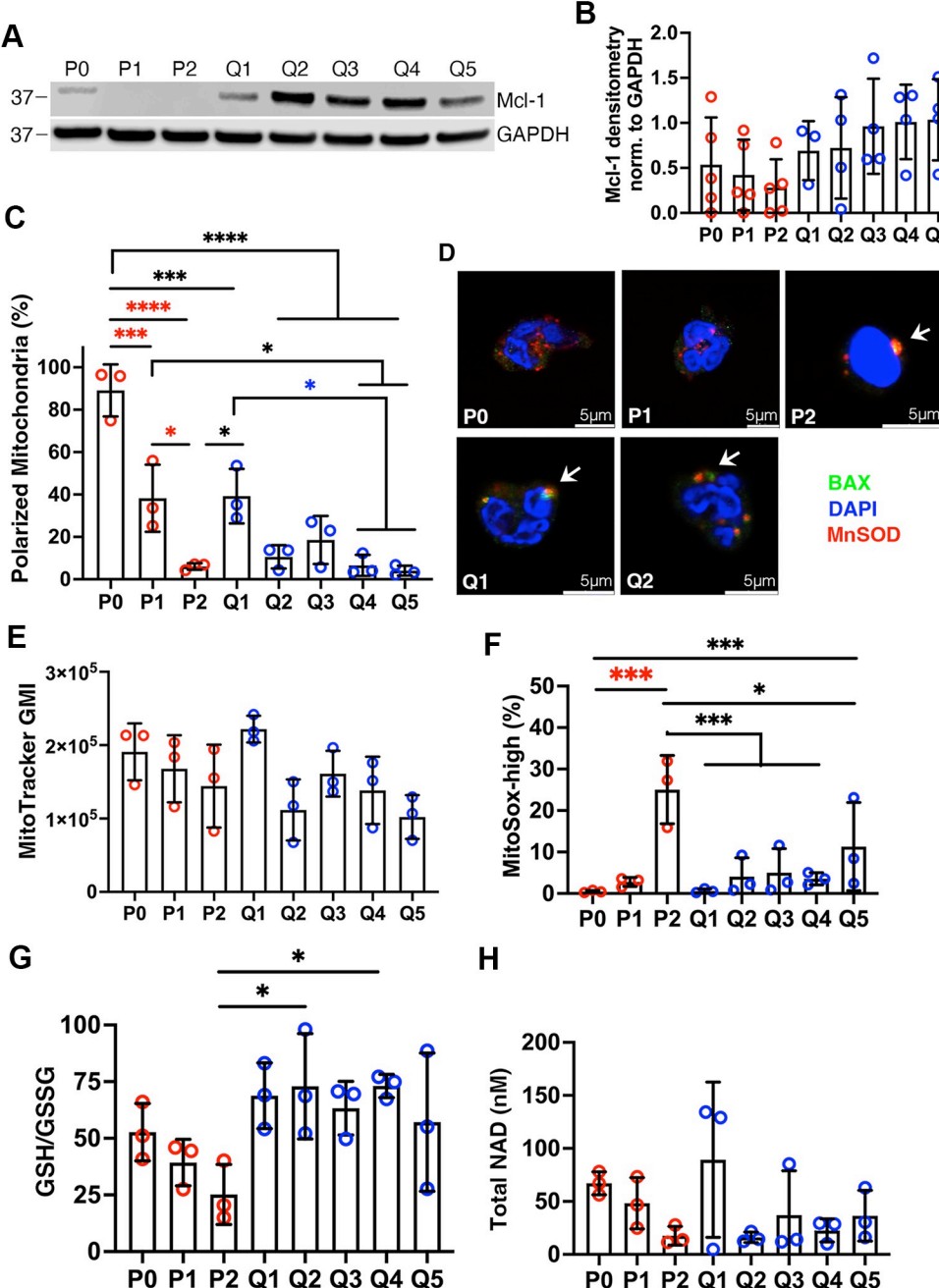

**Fig 3. Differential effects of QVD on mitochondrial integrity, abundance and oxidative stress.** Freshly isolated (P0), 1–2 day aged (P1, P2) and 1–5 day QVD-treated (Q1-Q5) neutrophils were analyzed. **(A-B)** MCL-1 in neutrophil lysates prepared at the indicated time points was detected by immunoblotting with GAPDH as the loading control. Representative immunoblots are shown in (A) with densitometric quantitation from four independent experiments in (B). **(C)** PMNs were stained with MitoProbe™ JC-1 and analyzed via flow cytometry at each time point. Data indicate the percentage of polarized mitochondria. **(D)** QVD does not prevent BAX translocation to mitochondria. Representative confocal images of control and QVD-treated PMNs were stained to detect BAX (green), the mitochondrial marker MnSOD (red) and DNA (blue). **(E)** PMNs were stained with MitoTracker™ Deep Red to quantify total mitochondrial mass as geometric mean intensity (GMI). **(F)** Neutrophils were stained with MitoSOX™ Red and analyzed by flow cytometry at each time point. The percentage of superoxide-positive mitochondria is shown. **(G)** Ratios of glutathione (GSH) to glutathione disulfide (GSSG) were quantified using GSH/GSSG-Glo™ Assay kits. **(H)** Quantitation of total neutrophil NAD. For graphs in C and E-H, data are the mean $\pm$ SD of three independent determinations and data were analyzed by two-way ANOVA with Tukey's multiple comparisons post-test. *$p < 0.05$, **$p < 0.01$, ***$p < 0.001$ and ****$p < 0.0001$.

**QVD treatment protects against mitochondrial oxidative stress.** Published data indicate that mitochondrial superoxide increases during apoptosis, demonstrating additional aspects of organelle dysfunction that accompany cell death [36, 37]. Using MitoSox™ staining and flow cytometry, we demonstrate that mitochondrial superoxide increased markedly in PMN aged for 48 h *in vitro* (P2) but not at earlier time points examined and was not elevated above baseline in QVD-treated cells until day 5 (Q5) (Fig 3F). Glutathione is an abundant antioxidant in neutrophils. During apoptosis, the ratio of reduced to oxidized glutathione (GSH/GSSG) declines, potentially contributing to increased oxidative stress within dying cells [37–39]. We now show that GSH/GSSG ratios remained high in QVD-treated cells over the five day time course examined yet were significantly lower in control PMNs by 48 h (P2) (Fig 3G). On the other hand, we confirmed that total NAD levels are low in neutrophils [37, 40, 41] and we show that this metabolite did not differ significantly in fresh, aged or QVD-treated cells (Fig 3H). Taken together, these data indicate that QVD sustains the ability of neutrophils to combat oxidative stress, including mitochondrial superoxide.

**Q-VD-OPh treatment sustains neutrophil functional capacity.** Having established that QVD-treated neutrophils remained viable for 5 days, our next objective was to determine the extent to which cell capacity for chemotaxis, phagocytosis, ROS production and degranulation were also sustained. As a first approach, we used an EZ-TAXIScan™ imaging system to generate movies of neutrophils migrating in an fMLF gradient and then quantified both the speed (instantaneous velocity) and directionality (chemotactic index) of individual cells using established procedures [23, 26]. By this assay, PMN capacity for rapid and directed migration was retained through 48 h and then declined precipitously, as the few aged cells that remained at 72 h were entirely nonmotile (Fig 4A and 4B). In sharp contrast, QVD-treated cells retained the capacity to chemotax through day 5, though migration speed and directionality were altered to different extents on days 3–5 (Q3-Q5) (Fig 4A and 4B). Specifically, the data indicate that average migration speeds (instantaneous velocities) were significantly slower on QVD days 3–4 whereas directionality (chemotactic indices), which indicates movement toward fMLF/total path length, was unimpaired until QVD day 5.

To assess phagocytosis, we fed cells opsonized yeast zymosan particles (OpZ) and quantified both the percentage of cells that ingested OpZ and the number of particles in each phagocytic cell. Representative light microscopy images are shown in Fig 5A, and pooled data are shown in Fig 5B and 5C. Our data demonstrate that neutrophil capacity for phagocytosis declined sharply between 24 and 48 h after isolation (P1-P2) yet was maintained at levels and efficiencies seen in freshly isolated cells for the 5 day duration of QVD treatment (Q1-Q5). The images in Fig 5A also confirm the nuclear morphology data shown in Fig 1A.

Next, we used the luminol chemiluminescence assay to evaluate ROS production by neutrophils following stimulation with 200 nM PMA [42]. The kinetics and magnitude of PMA-stimulated oxidant production by control and QVD-treated cells assayed 5 h after isolation were nearly identical and neither set of cells produced NADPH oxidase-derived ROS prior to stimulation (Fig 6A). The PMA-stimulated respiratory burst was significantly diminished in aged PMNs by day 2, as indicated by a slower rate of respiratory burst onset and an $85 \pm 5.2\%$ reduction in peak oxidant production (Fig 6A and 6B). By contrast, PMA-stimulated ROS produced by QVD-treated cells resembled fresh cells through day 2 and was diminished by 47–50% but not ablated between day 3 and day 5 (Q3, Q5) (Fig 6A and 6B).

Mobilization of intracellular granules also contributes to microbe killing by neutrophils [2]. To assess capacity for degranulation, we quantified release of elastase (ELA2) into the extracellular medium using ELISA (Fig 7). By this assay, the extent of elastase release by fresh, aged or QVD-treated cells did not differ significantly despite upward trends for aged control and QVD-treated cells.

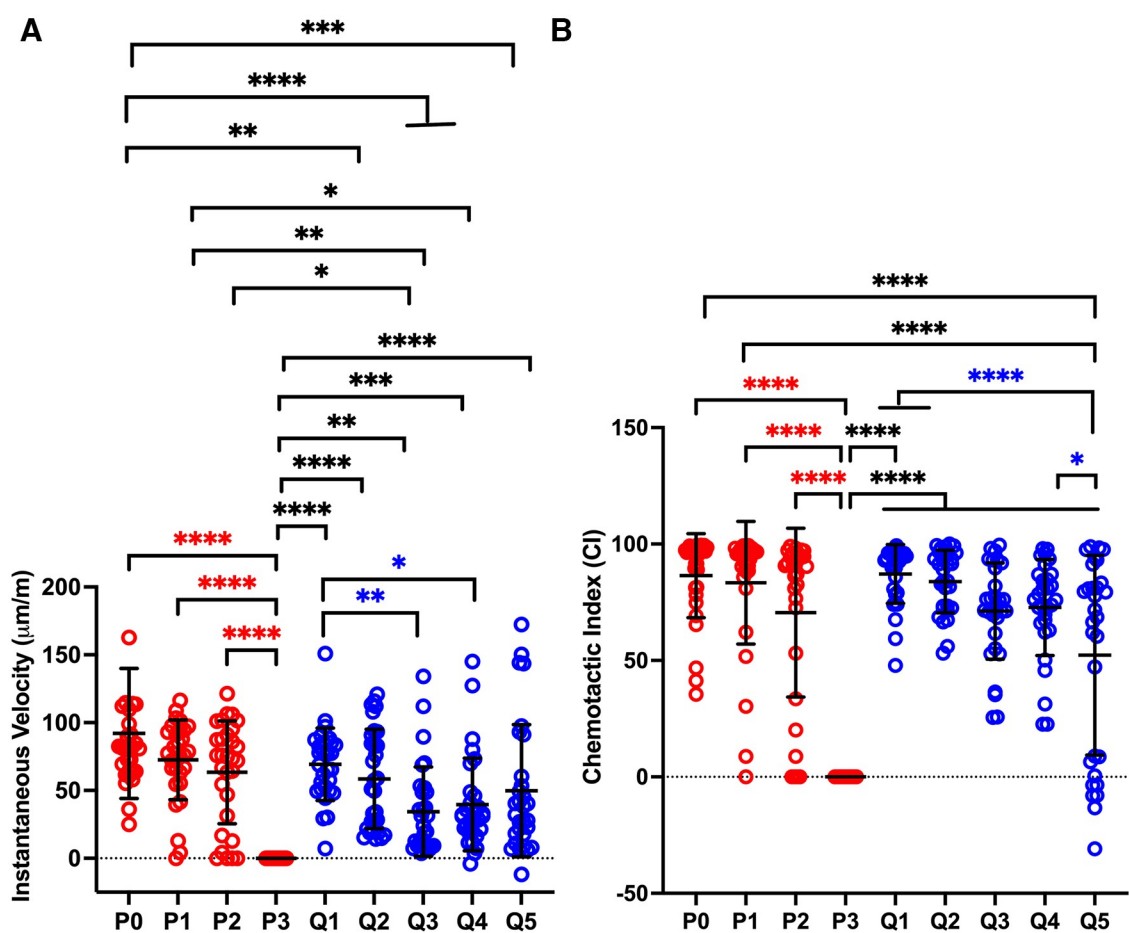

**Fig 4. QVD preserves neutrophil chemotaxis to fMLF.** Chemotaxis of fresh (P0), 1–3 day aged (P1-P3) control and 1–5 day QVD-treated (Q1-Q5) PMNs migrating in an fMLF gradient was analyzed with EZ-TAXIScan™ video imaging. Instantaneous velocities (**A**) and chemotactic indices (**B**) of individual cells are shown. Data are the mean ± SD of three independent experiments. For all graphs data were analyzed by two-way ANOVA with Tukey's multiple comparisons post-test. *$p < 0.05$, **$p < 0.01$ ***$p < 0.001$ and ****$p < 0.0001$.

Taken together, these data demonstrate that QVD-treated neutrophils retained fundamental defense functions for at least 5 days in culture, though reduced efficiency of ROS production and chemotaxis were apparent beginning on QVD day 3. By contrast, 48 h aged neutrophils could migrate rapidly toward fMLF and degranulate, but their capacity to ingest OpZ and generate NADPH oxidase-derived ROS were nearly ablated.

**Q-VD-OPh has limited effect on neutrophil surface markers.** Mature neutrophils display CD15, CD16 and CD62L at the cell surface and we quantified these molecules on fresh, aged, and QVD-treated neutrophils using reagents in DURAclone IM granulocyte kits followed by flow cytometry analysis (S5 Fig). CD15, also called Lewis X, is a human myeloid cell marker [43]. CD15 levels were somewhat variable but did not differ significantly on freshly isolated, aged or QVD-treated cells.

In previous work, we confirmed that mature neutrophils are CD62L^bright/CD16^bright [44]. CD62L declines progressively as cells age in culture or upon activation, whereas CD16 disappears during apoptosis but can also be diminished on live cells if surface shedding exceeds rates of *de novo* synthesis and secretion [45–48]. As expected, CD62L was nearly undetectable

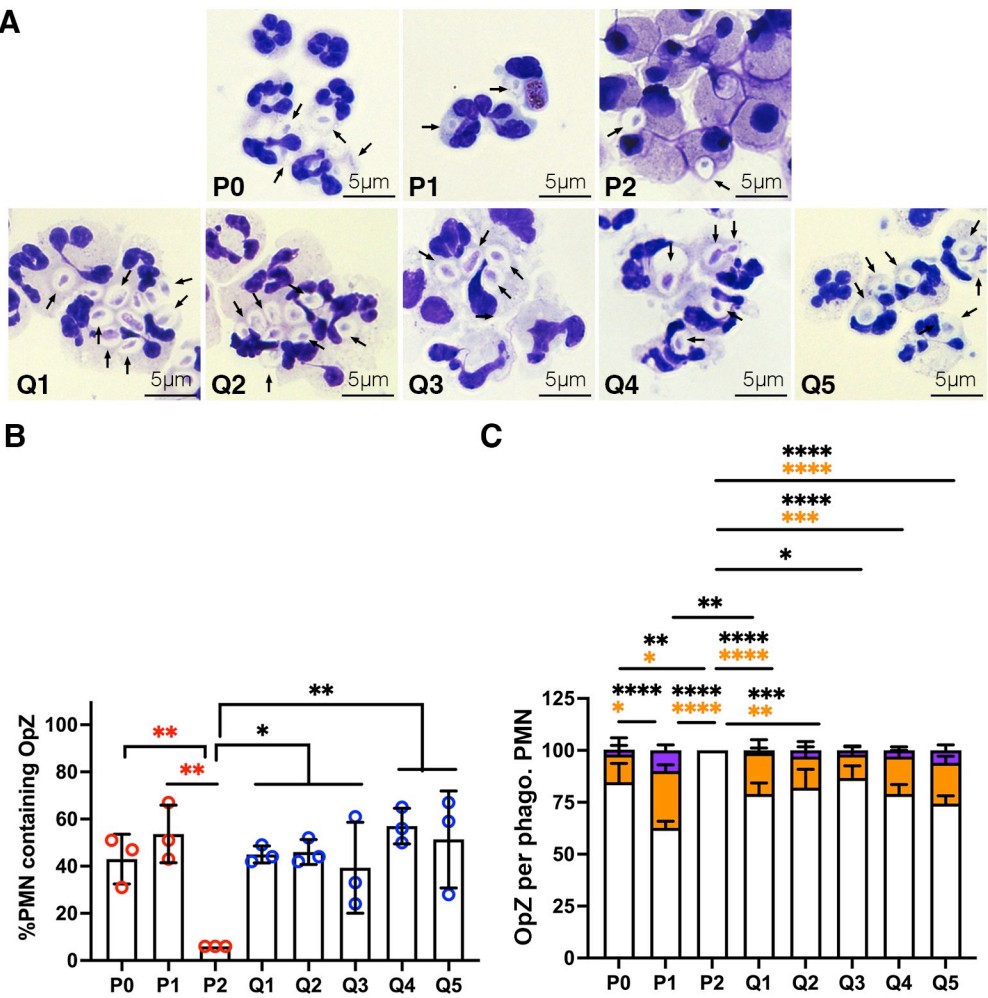

**Fig 5. Phagocytic capacity of QVD-treated PMNs is unchanged over 5 days.** At each time point, fresh, aged and QVD-treated neutrophils were fed OpZ particles at a ratio of 4 particles per cell and incubated for 15 min at 37°C with nutation. (**A**) Representative images of Hema-3-stained control (P0-P2) and QVD-treated (Q1-Q5) PMNs at each time point. Arrows indicate OpZ phagosomes. (**B**) Quantitation of the percentage of PMNs that ingested OpZ at each time point. Data are the mean $\pm$ SD of three independent experiments. (**C**) Quantitation of the number particles in each cell that ingested OpZ. White bars 1–2 OpZ/cell, orange bars 3–4 OpZ/cell, purple bars 5–6 OpZ/cell. For all graphs data were analyzed by two-way ANOVA with Tukey's multiple comparisons post-test. *$p < 0.05$, **$p < 0.01$, ***$p < 0.001$ and ****$p < 0.0001$.

on both aged and QVD-treated cells by 24 h. On the other hand, as compared with aged PMNs, QVD treatment slowed changes in surface CD16, reinforcing the fact that shedding of this surface marker can be uncoupled from apoptosis (S5 Fig).

CD11b is present at low levels on resting PMNs, is upregulated by cell activation, and plays important roles in cell adhesion, migration and phagocytosis [48]. In keeping with the role of complement receptor 3 (CD11b/CD18 heterodimers) in OpZ phagocytosis [49] and the data in Fig 5, our flow cytometry data indicate that CD11b levels were sustained by QVD over the 5-day assay period and were generally higher on QVD-treated cells than on control cells that had been aged for 48 h (P2) (S5 Fig).

**Autophagy is not significantly altered by Q-VD-OPh.** In neutrophils, autophagy plays a role in regulation of degranulation and NADPH oxidase activity but not phagocytosis or cell

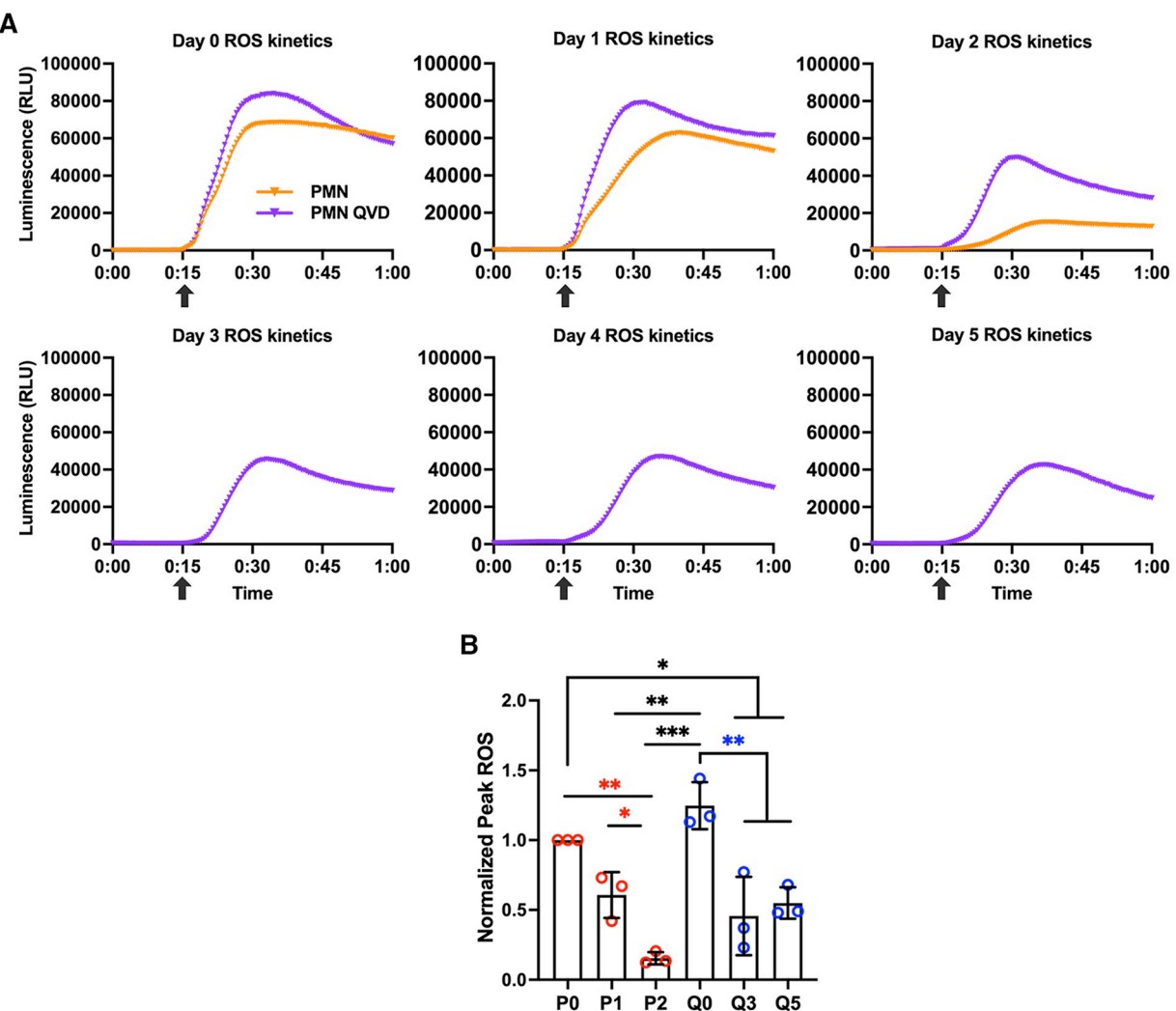

**Fig 6. Respiratory burst capacity is diminished but not absent 3–5 days after QVD treatment.** Oxidant production was detected using the luminol chemiluminescence assay. Responses were measured every 30 sec for 60 min and PMA was added to 200 nM final concentration at 15 min (*arrows*). (**A**) Representative data obtained for control and QVD-treated cells on days 0–5 as indicated. Data points are the mean of triplicate technical replicates. (**B**) Pooled data from three independent experiments show peak ROS normalized to freshly isolated control cells (P0) stimulated with PMA. Data are the mean ± SD of three independent experiments and were analyzed by Student's two-tailed *t*-test. *$p < 0.05$, **$p < 0.01$ and ***$p < 0.001$.

migration [50]. Links between autophagy and apoptosis have been described but are context-specific, as autophagy can increase or decrease apoptosis or occur in parallel without being linked mechanistically [51, 52].To assess possible effects of Q-VD-OPh on autophagy in neutrophils, we quantified the autophagy regulator mTOR, its phosphorylation on S2448 [53], and the autophagosome markers LC3B-II and p62 (also called sequestrome 1, SQSTM1) using flow cytometry, western blotting and confocal microscopy, respectively (S6 Fig). Our data indicate a trend toward reduced total and S2446-phosphorylated mTOR in 48 h-aged PMNs (P2) relative to other conditions tested, but this was not statistically significant and ratios of total to phosphorylated mTOR did not significantly differ for fresh, aged or QVD-treated PMNs at any assayed time point (S6A Fig). LC3B-II plays a role in autophagosome formation and conversion of LC3B-I to LC3B-II can indicate autophagy initiation [54, 55]. In our hands, levels of

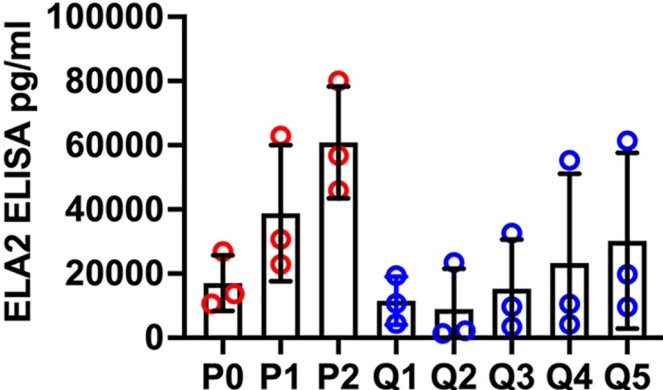

**Fig 7. Quantitation of elastase release.** Release of elastase (ELA2) into the culture medium by fresh (P0), 24-48h aged (P1-P2) or 1–5 day QVD-treated (Q1-Q5) cells was quantified by ELISA. Data are the mean ± SD of three independent experiments and were analyzed by two-way ANOVA with Tukey's multiple comparisons post-test.

LC3B-I and to a lesser extent LC3B-II were variable across independent experiments and no consistent pattern could be discerned (S6B Fig). P62 is an autophagosome marker that tethers cargo to LC3B-II in the autophagosome membrane [54]. Confocal analysis showed that neutrophils contained relatively few autophagosomes (S6C Fig, *arrows*), as fresh and 24 h-aged cells generally contained no p62 puncta, and cells aged for longer periods of time generally contained 5 or fewer puncta, regardless of QVD treatment. Addition of bafilomycin at 24 h (P1, Q1) to block autophagosome-lysosome fusion had no discernable effect on autophagosome abundance when cells were analyzed 24 h later (P2, Q2). Moreover, diffuse staining for p62 increased selectively throughout the cytosol of 48 h-aged control PMNs (P2). Considered together, these data suggest that autophagy levels are relatively low and do not significantly differ in fresh (P0), aged (P1, P2) and QVD-treated (Q1-Q5) PMNs.

## Discussion

The central finding of this study is that a single dose of QVD is sufficient to prevent apoptosis of human neutrophils for at least 5 days *in vitro* as indicated by measurements of caspase activity and processing, externalization of PS, DNA fragmentation and acquisition of an apoptotic nuclear morphology. We also show that QVD prevented the increases in mitochondrial ROS and the decline of GSH/GSSG ratios that are known to accompany apoptosis, whereas total NAD was unchanged, in keeping with the limited role of mitochondria in PMN metabolism and the greater role of GSH in protection against oxidative stress [39, 40, 56, 57]. At the same time, we also show that QVD sustains cell capacity for phagocytosis, chemotaxis, ROS production and degranulation, albeit to different extents. Thus, our data confirm and extend the results of previous studies which demonstrated the ability of QVD to preserve neutrophil viability and function for up to 48 hours, the longest time point examined [16, 18, 19].

An unexpected finding of this study was the progressive decline of procaspase-3 in lysates of neutrophils during prolonged QVD treatment shown in Fig 2, and these data are noteworthy for two reasons. On the one hand, they demonstrate constitutive synthesis of procaspase-3 in neutrophils that is sensitive to CHX, as has been shown in other cell types [58]. On the other hand, the MG-132 and bortezomib data revealed proteasomal degradation of procaspase-3 resulting from extended QVD treatment with concomitant appearance of a ~20kDa band in cell lysates. Based on these data, we favor a model in which QVD-stimulated procaspase-3

degradation exceeds the rate of *de novo* synthesis, leading to its depletion from neutrophil lysates. Although the precise nature of the ~20 kDa fragment remains to be determined, these data advance understanding of apoptosis inhibition by QVD.

In addition to apoptosis, pyroptosis, necroptosis, NETosis and autophagy are other mechanisms that can directly mediate or facilitate neutrophil death [2]. Pan-caspase inhibition is sufficient to prevent apoptosis and pyroptosis, but the fate of QVD-treated neutrophils is unknown. Macrophages treated with pan-caspase inhibitors undergo rapid receptor-interacting serine/threonine protein kinase (RIPK)-mediated necroptosis upon exposure to Tumor Necrosis Factor-alpha (TNFα) [59]. However, *in vitro* and *in vivo* studies demonstrate that QVD does not sensitize human or murine neutrophils to necroptosis. Rather, these cells remain viable in the presence of TNFα unless XIAP is also deleted, and at the molecular level XIAP has been shown to target RIPK1 for degradation [59–61]. Thus, sustained expression of XIAP in QVD-treated neutrophils (S3 Fig) likely prevented necroptosis under the conditions of this study. At the same time, we did not detect evidence of nuclear membrane dissolution or DNA release in our light microscopy analyses of Hema-3-stained neutrophils through day 5, excluding a role for NETosis in cell demise.

Factors in the hepatocellular carcinoma microenvironment induce autophagy of human neutrophils and delay apoptosis onset that is associated with enhanced mitochondrial stability and reduced organelle dysfunction [62]. The confocal images in S6C Fig show that autophagosomes are rare or absent in freshly isolated neutrophils in normal media, confirming published data [62], but at the same time, our analysis of p62, LC3-II and mTOR (S6 Fig) suggest that neither constitutive apoptosis of aged PMNs nor the extended survival of cells treated with QVD for up to 5 days is linked to prominent changes in autophagy. Inasmuch as subtle changes in autophagy cannot be excluded, additional studies are also needed to determine if autophagy accounts for the cytoplasmic vacuolation that was apparent from QVD day 6 onward (S1 Fig) and if nutrient depletion contributes to cell demise.

Neutrophils contain fewer mitochondria than other cell types [31, 63]. Nonetheless, these organelles contribute to cell metabolism and play important roles during apoptosis and chemotaxis [63]. Key early events in the intrinsic apoptosis pathway are MOMP and release of cytochrome *c* from the intermembrane space into the cytosol, which initiates formation of the apoptosome for processing and activation of procaspase-9 [1, 2]. In healthy neutrophils, integrity of the OMM is maintained by MCL-1 and to a lesser extent A1, which bind and sequester the pore-forming proteins BAX and BAK (BCL-2 antagonist/killer) in the cytosol [1]. As MCL-1 has a half-life of only 3 hours, its continued synthesis is essential for neutrophil survival [64, 65].

JC-1 is a dye that enters mitochondria and emits bright red fluorescence in organelles with high membrane potential and green fluorescence in organelles with low membrane potential [24, 66]. For this reason, JC-1 staining is commonly used as an indicator of MOMP during apoptosis [1, 24, 66]. We used this assay to demonstrate that QVD did not prevent MOMP despite the presence of MCL-1 in cell lysates and conclude that although MCL-1 is critical for neutrophil survival it is not sufficient. At the molecular level, this likely reflects the fact that MCL-1 regulation is multifactorial and is influenced by its interactions with multiple binding partners in addition to phosphorylation and ubiquitination [31, 67, 68]. For example, BIM (BCL-2-interacting mediator of cell death) and PUMA (p53-upregulared modulator of apoptosis) can bind to and stabilize MCL-1 while also displacing BAX and BAK, thereby enabling MOMP [69], and we speculate that this mechanism may be operative QVD-treated PMNs. Of note, delayed apoptosis and sustained abundance of MCL-1 and XIAP despite MOMP are also characteristics of neutrophils treated with G-CSF [12].

Mitochondrial dynamics are understudied in neutrophils. However, a recent study that combined JC-1 staining with live-cell imaging suggests that mitochondrial membrane potential may be more dynamic and heterogeneous than previously appreciated [70]. It is established that during chemotaxis translocation of mitochondria to the leading edge is essential for cell motility, and in this locale mitochondria release ATP that gains access to the extracellular space and engages P2Y2 purinergic receptors initiating autocrine signaling and local actin polymerization [63, 70]. Live cell imaging revealed that when neutrophils were placed in an fMLF gradient, a subset of mitochondria that emit bright red JC-1 fluorescence translocate to the leading edge whereas most mitochondria remain at the cell rear and emit green fluorescence [70]. Additional data showed that whether mitochondria emitted red or green fluorescence was dynamically regulated by P2Y2 signaling at the front of the cell and adenosine A2a receptor signaling at the rear [70]. Notably, deletion of P2Y2 diminished mitochondrial membrane potential throughout the entire cell, whereas deletion of A2a had the opposite effect. Taken together, these data suggest that mitochondrial membrane potential is dynamically regulated in live neutrophils. Moreover, complete dissipation of mitochondrial membrane potential using uncoupling agents renders cells entirely nonmotile but only slightly diminishes other functions, such as NADPH oxidase activity [66, 70]. Thus, the fact that a majority of aged neutrophils in our EZ-TAXIScan™ assays were able to migrate in fMLF gradients over 48 hours indicates that critical mitochondrial functions were maintained despite apoptosis onset (Fig 4), in agreement with published data [5]. In addition, we show that chemotaxis of QVD-treated neutrophils was sustained but somewhat altered, as speed of migration toward fMLF was diminished on days 3–4, whereas directionality was unaffected until day 5 (Fig 4). At the same time, total mitochondrial mass was unchanged (Fig 3E). Precisely what accounts for these changes remains to be determined, but we speculate that Rac, Rho kinase and myosin II signaling, which regulate the speed and efficiency of neutrophil amoeboid movement, and chemotaxis receptor signaling, which regulates gradient sensing and directionality of migration, are differentially effected [23], and the available data set the stage for future in-depth studies of mitochondrial dynamics and function. With future studies in mind, it is noteworthy that studies in several cell types show that mitochondrial abundance can fluctuate in response to metabolic demand and that damaged organelles can either be repaired, expelled from cells, degraded by mitophagy or replaced by mitogenesis [33, 35, 63].

Regarding other host defense functions, degranulation was not significantly altered, but both phagocytosis and NADPH oxidase activity declined in neutrophils aged for more than 24 hours in culture (Figs 5–7), in agreement with published data [5, 7, 71]. Conversely, phagocytic capacity was largely unchanged in neutrophils treated with QVD, whereas PMA-stimulated oxidant production was reduced by ~50% on days 3–5. To our knowledge, the underlying molecular mechanisms are unknown, but could potentially include changes in the abundance of one or more NADPH oxidase subunits, reduced availability of enzyme cofactors, or changes in the signaling pathways that control the kinetics of enzyme assembly and activation at the membrane [72].

Recently, Fan *et al.*, reported that a complex drug cocktail comprised of 50 μM QVD, 1 μM deferoxamine, 10 pM Heat-shock protein 70, 10 μM N-acetyl cysteine, 10 μM Necrostatin 1s, and 10 ng/ml G-CSF prolonged the viability of murine and human neutrophils such that 50% and 20% of cells, respectively, remained healthy on day 7 [73]. Assays of cell function showed that phagocytosis was sustained for 5 days in the drug-treated human cells, whereas rates of migration toward fMLF, but not directionality, were significantly reduced on day 3 and day 5, and oxidant production was diminished by ~25%. These data resemble the results of this study and lend further support to the conclusion that QVD is the primary factor in their drug cocktail that sustains neutrophil viability and function. On the other hand, Fan *et al.*, also show that

phagocytosis, chemotaxis and ROS production by aged, untreated human neutrophil controls declined more rapidly than we report here, as all these functions were significantly impaired within 6–12 hours of cell isolation. This discordance is likely due to methodology differences that affect cell age, as we immediately processed fresh blood samples to isolate neutrophils whereas Fan *et al.* obtained leukocytes from filters being discarded by the blood bank, which extended the interval between blood collection and neutrophil isolation by an unknown amount of time.

Dysregulated apoptosis of neutrophils and other cell types is a feature of many human diseases. The potency of QVD together with its ability to cross the blood-brain barrier and its lack of *in vivo* toxicity make it an attractive adjunctive therapy for several diseases including but not limited to ischemic stroke, spinal cord injury, and reperfusion injury [29], and studies in mice demonstrate significant benefit of QVD as a host-directed therapy for treatment of methicillin-resistant *Staphylococcus aureus* skin infections [59]. In addition, the inherently short lifespan of PMNs is a significant limitation to research, as methodologies that require extended incubation periods, such as transfection, are not feasible. There are no cell lines that mimic mature human neutrophils and no model similar to the Hoxb8 system that allows genetic manipulation of murine neutrophil precursors prior to differentiation [74]. Thus, it is conceivable that QVD-treated neutrophils might be useful as a tool to advance studies of human neutrophil biology.

In summary, the results of this study significantly advance understanding of the extent to which pan-caspase inhibition by QVD can extend the functional lifespan of human neutrophils. Our data confirm links between mitochondrial membrane potential and chemotaxis. At the same time, we identified new consequences of prolonged QVD treatment, including downregulation and disappearance of procaspase-3 and extensive cell vacuolation that merit further study. The opposing effects of QVD on the abundance of procaspase-3, XIAP and MCL-1 are of particular interest as they suggest that the mechanisms of crosstalk between apoptotic caspases and the changes in gene expression that control the balance of pro- and anti-apoptotic regulatory factors in the cytosol are incompletely defined.

## Supporting information

**S1 Fig. Morphology of human neutrophils after prolonged QVD treatment.** Representative Hema-3 images of human neutrophils that were incubated in medium containing 10 μM QVD for 6–10 days (Q6-Q10) and then stained with Hema-3 reagents. *Arrows* indicate cytoplasmic vacuoles.
(TIF)

**S2 Fig. Lower concentrations of QVD do not confer sustained apoptosis inhibition.** Freshly isolated neutrophils were aged in culture in the presence and absence of 0.1, 1 or 10 μM QVD, as indicated, prior to analysis. **(A)** Graphs show the percentage of healthy cells for each condition and time point as determined by Annexin V-FITC/PI staining and flow cytometry. Data are the average + SD from three independent experiments for days 1 and 3 and 2–3 experiments for day 5. Data from days 1 and 3 were analyzed by one-way ANOVA with Tukey's multiple comparisons post-test. $*p<0.05$, $**p<0.01$. **(B)** Representative images of Hema-3-stained cells show the nuclear morphology of cells treated with the indicated concentrations of QVD for 24 or 72 h.
(TIF)

**S3 Fig. Full-length XIAP is present in QVD-treated neutrophils.** Immunoblot of neutrophil lysates prepared on days 0–2 (P0-P2) or after 1–5 days of QVD treatment (Q1-Q5) probed to

detect XIAP with GAPDH as the loading control. A p30 XIAP fragment indicative of apoptosis was present in aged PMNs on days 1 and 2 but was absent in freshly isolated cells and cells treated with QVD for 1–5 days.
(TIF)

**S4 Fig. JC-1 flow cytometry dot plots and gating strategy.** Neutrophils were stained with MitoProbe™ JC-1 and analyzed via flow cytometry at each time point. P0-P2, freshly isolated, 24 h and 48 h-aged neutrophils, respectively. Q1-Q5, neutrophils treated for 1–5 days with Q-VD-OPh.
(TIF)

**S5 Fig. Surface markers of control and QVD-treated neutrophils.** Freshly isolated neutrophils (P0), cells that were cultured in medium for 1–2 days (P1, P2) and cells that were incubated for up to 5 days in medium containing 10 μM QVD (Q1-Q5) were stained using the DURAClone IM Granulocytes kit and analyzed via flow cytometry. Pooled data from three independent experiments show the average geometric mean intensity (GMI) $\pm$ SD for CD15, CD62L, CD16 and CD11b, as indicated. For all graphs data were analyzed by two-way ANOVA with Tukey's multiple comparisons post-test. $^*p<0.05$ and $^{**}p<0.01$.
(TIF)

**S6 Fig. Effects of aging and QVD on autophagy markers.** Freshly isolated (P0), 24–48 h aged cells (P1-P2) and cells treated for 1–5 days with Q-VD-OPh (Q1-Q5) were compared. **(A)** Graphs show the total mTOR, S2446-phosphorylated mTOR and the ratio of total/phosphorylated mTOR for each condition. Data are the mean ± SD of three independent experiments. **(B)** Western blots of cell lysates were probed to detect LC3B-I and LC3B-II with GAPDH as the loading control. Data for three independent experiments are shown. **(C)** Representative confocal images show p62 in green and nuclear DNA (DAPI) in blue. *Arrows* indicate p62-positive autophagosomes.
(TIF)

**S1 Raw images. Uncropped immunoblots.**
(PDF)

## Author Contributions

**Conceptualization:** Lee-Ann H. Allen.

**Formal analysis:** Lisa Khuu, Alisha Pillay, Allan Prichard, Lee-Ann H. Allen.

**Funding acquisition:** Lee-Ann H. Allen.

**Investigation:** Lisa Khuu, Alisha Pillay, Allan Prichard.

**Project administration:** Lee-Ann H. Allen.

**Resources:** Lee-Ann H. Allen.

**Supervision:** Lee-Ann H. Allen.

**Visualization:** Lisa Khuu, Alisha Pillay, Lee-Ann H. Allen.

**Writing – original draft:** Lisa Khuu, Lee-Ann H. Allen.

**Writing – review & editing:** Lisa Khuu, Lee-Ann H. Allen.

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
