## [Decision Letter · Decision Letter 0]

17 Oct 2024

PONE-D-24-41135Effects of the pan-caspase inhibitor Q-VD-OPh on human neutrophil lifespan and functionPLOS ONE

Dear Dr. Allen,

Thank you for submitting your manuscript to PLOS ONE. After careful consideration, we feel that it has merit but does not fully meet PLOS ONE’s publication criteria as it currently stands. Therefore, we invite you to submit a revised version of the manuscript that addresses the points raised during the review process.

Dear Authors,Please provide point by point answers to the all the reviewers comments. 

We look forward to receiving your revised manuscript.

Kind regards,

Saeid Ghavami, PhD

Academic Editor

PLOS ONE

“This work was supported, in part, by funds from the US government, including National Institutes of Health R01 AI119965 awarded to L-AA. Part of this study was also supported by funds awarded to L-AA by the University of Missouri School of Medicine.”

“This work was supported, in part, by funds from the US government, including National Institutes of Health R01 AI119965 awarded to L-AA. Part of this study was also supported by funds awarded to L-AA by the University of Missouri School of Medicine.”

“This work was supported, in part, by funds from the US government, including National Institutes of Health R01 AI119965 awarded to L-AA. Part of this study was also supported by funds awarded to L-AA by the University of Missouri School of Medicine.”

4. Please update your submission to use the PLOS LaTeX template. The template and more information on our requirements for LaTeX submissions can be found at http://journals.plos.org/plosone/s/latex.

7. PLOS ONE now requires that authors provide the original uncropped and unadjusted images underlying all blot or gel results reported in a submission’s figures or Supporting Information files. This policy and the journal’s other requirements for blot/gel reporting and figure preparation are described in detail at https://journals.plos.org/plosone/s/figures#loc-blot-and-gel-reporting-requirements and https://journals.plos.org/plosone/s/figures#loc-preparing-figures-from-image-files. When you submit your revised manuscript, please ensure that your figures adhere fully to these guidelines and provide the original underlying images for all blot or gel data reported in your submission. See the following link for instructions on providing the original image data: https://journals.plos.org/plosone/s/figures#loc-original-images-for-blots-and-gels.  

Reviewers' comments:

Reviewer's Responses to Questions

**Comments to the Author**

1. Is the manuscript technically sound, and do the data support the conclusions?

Reviewer #1: Yes

Reviewer #2: Yes

2. Has the statistical analysis been performed appropriately and rigorously? 

Reviewer #1: Yes

Reviewer #2: I Don't Know

3. Have the authors made all data underlying the findings in their manuscript fully available?

Reviewer #1: Yes

Reviewer #2: Yes

4. Is the manuscript presented in an intelligible fashion and written in standard English?

Reviewer #1: Yes

Reviewer #2: Yes

5. Review Comments to the Author

Reviewer #1: The manuscript by Lee-Ann Allen et al. is a significant contribution to our understanding of neutrophil survival and function. The authors investigate the potential of the pan-caspase inhibitor Q-VD-OPh to extend neutrophil lifespan. Their findings are intriguing and demonstrate a significant extension of neutrophil lifespan and maintenance of several cell functions, such as viability and phagocytosis (with reduced efficiency of ROS production, chemotaxis and instantaneous velocity from day 3). The experiments are well-designed and solid.

Some points listed below must be addressed to enhance the manuscript further.

1- All full names must be stated before using the abbreviation (for ex. MCL-1, BAX, BCL-2).

2- Line 40_Introducing a few words about NETosis' characteristics would help the reader.

3- In materials and methods, providing a brief description of the process of neutrophil isolation would enhance the reader's understanding and provide the necessary context.

4- Please include the statistical tests used in all figure legends (Fig.1, 2, 3, 4, 5 and S4).

5- Fig 2 B, C, D, E, F, G: Adding the word days to the x-axis will improve the comprehension of the figure. Although the results are strong, the order of the graph in this figure is confusing; please reorganize it for better visualization. In the legend, it would also be helpful to have the word 1-5 days in QVD-treated neutrophils (Q1-Q5).

6- Fig 3. The order graph is confusing; please reorganize it for better visualization.

7- Please please add the size bar information in all micrographs.

8- Supplementary Fig. 1: Please specify the abbreviation used in the graph: (Q6-Q10).

9-In the results, please state and further explain the changes in instantaneous velocity and chemotactic index. It would also be beneficial to discuss the implications of these findings specifically for the bigger picture.

Reviewer #2: 1) Synopsis:

The manuscript investigates the effects of the pan-caspase inhibitor Q-VD-OPh on human neutrophil lifespan and functionality. The authors show that Q-VD-OPh significantly extends neutrophil viability by preventing apoptosis for five days while maintaining functional capacities such as phagocytosis and chemotaxis. However, mitochondrial depolarization still occurs despite the inhibition of apoptosis.

2) Strengths:

Comprehensive apoptosis inhibition analysis: The study uses multiple assays to confirm the inhibition of apoptosis, providing robust evidence.

Functional focus: The paper evaluates not only apoptosis inhibition but also crucial neutrophil functions, ensuring a more comprehensive understanding of cell survival.

Clinical relevance: Extending neutrophil lifespan has potential therapeutic implications in conditions where neutrophil function is compromised.

3) Weaknesses:

Incomplete mechanistic insight: While the inhibition of caspases is well demonstrated, the underlying molecular mechanisms explaining the preservation of function despite mitochondrial dysfunction need further elaboration.

Lack of in vivo validation: All experiments are performed in vitro, limiting the translatability of the findings to real-life physiological scenarios.

Neutrophil viability versus functionality: The study highlights that while apoptosis is inhibited, mitochondrial depolarization still occurs, raising questions about the long-term viability and effectiveness of these cells in a physiological context.

4) Critical Comments for Major Revision:

1) Impact of Inhibitor on Mitochondrial Function:

To evaluate the effect of Q-VD-OPh on mitochondrial function, focus on mitochondrial membrane potential, ROS production, and redox balance. Use JC-1 staining to assess membrane potential and analyze changes in mitochondrial depolarization. For ROS, employ both flow cytometry (DHE or MitoSOX) and luminol-based assays to detect superoxide production. Additionally, measure mitochondrial redox state by monitoring NADH/NAD+ ratios or GSH/GSSG ratios, which will indicate oxidative stress levels and mitochondrial health.

2) Inhibitor's Role in Cellular Pathways Like Autophagy:

To explore whether Q-VD-OPh affects autophagy, perform western blotting to detect autophagy markers such as LC3-II, p62, and Beclin-1. Combine this with transmission electron microscopy (TEM) to visualize autophagosomes. Use inhibitors like chloroquine or bafilomycin A1 to evaluate autophagic flux in combination with Q-VD-OPh. Additionally, assess the activity of mTOR (key autophagy regulator) via its phosphorylation status using phospho-specific antibodies. This will clarify whether Q-VD-OPh modulates autophagy to support neutrophil survival.

3) Effects on Neutrophil Natural Behavior:

Investigate if Q-VD-OPh alters the natural behavior of neutrophils by studying their chemotaxis, phagocytosis, and degranulation. Use EZ-TAXIScan assays to evaluate neutrophil migration towards fMLF gradients. For phagocytosis, opsonized zymosan assays combined with fluorescence microscopy will quantify particle uptake. Assess degranulation by measuring extracellular levels of enzymes like MPO or elastase using ELISA. These functional assays will reveal if Q-VD-OPh preserves normal neutrophil behavior or induces functional impairments.

These experimental approaches will provide a detailed understanding of how Q-VD-OPh impacts neutrophil survival, mitochondrial integrity, autophagy, and functionality.

6. PLOS authors have the option to publish the peer review history of their article (what does this mean?). If published, this will include your full peer review and any attached files.

Reviewer #1: No

Reviewer #2: **Yes: **Saeid Ghavami

---

## [Author Response · Author response to Decision Letter 0]

1 Dec 2024

Response to the Reviewers - Point-by-point 

We appreciate the generally positive comments of the Reviewers and the opportunity to revise our manuscript for possible publication in PLoS One. We believe that the changes made in response to the critiques were beneficial and improved the quality of our study.

Please ensure that your manuscript meets PLOS ONE's style requirements, including those for file naming. We followed all style and file naming requirements. 

Please state what role the funders took in the study. Funders had no role in this study. The statement - "The funders had no role in study design, data collection and analysis, decision to publish, or preparation of the manuscript" – is noted in the Cover Letter, as requested. 

We note that you have provided funding information that is currently declared in your Funding Statement. However, funding information should not appear in the Acknowledgments section or other areas of your manuscript. Funding information was removed from the Acknowledgements section of the manuscript and is now only in the Funding Statement.

Please update your submission to use the PLOS LaTeX template. The template and more information on our requirements for LaTeX submissions can be found at http://journals.plos.org/plosone/s/latex. We carefully followed the LaTeX template when creating the revised manuscript. 

Please amend either the abstract on the online submission form (via Edit Submission) or the abstract in the manuscript so that they are identical. – We ensured that the Abstract is identical in the Submission Form and the manuscript. 

Your ethics statement should only appear in the Methods section of your manuscript. If your ethics statement is written in any section besides the Methods, please delete it from any other section. – The Ethics Statement was deleted from the end of the manuscript and is now at the beginning of the Materials and Methods section (lines 65-71). 

PLOS ONE now requires that authors provide the original uncropped and unadjusted images underlying all blot or gel results reported in a submission’s figures or Supporting Information files. In your cover letter, please note whether your blot/gel image data are in Supporting Information or posted at a public data repository. – As with our first submission, we provided original, uncropped blots as Supporting Information. This is indicated in the Cover Letter.

Reviewer #1

The manuscript by Lee-Ann Allen et al. is a significant contribution to our understanding of neutrophil survival and function. The authors investigate the potential of the pan-caspase inhibitor Q-VD-OPh to extend neutrophil lifespan. Their findings are intriguing and demonstrate a significant extension of neutrophil lifespan and maintenance of several cell functions, such as viability and phagocytosis (with reduced efficiency of ROS production, chemotaxis and instantaneous velocity from day 3). The experiments are well-designed and solid.

We appreciate your positive comments regarding our manuscript. 

Some points listed below must be addressed to enhance the manuscript further.

1. All full names must be stated before using the abbreviation (for ex. MCL-1, BAX, BCL-2).

As requested, all abbreviations have been defined at first use throughout the document.

2. Line 40_Introducing a few words about NETosis' characteristics would help the reader.

As requested, additional characteristics of death mediated by release of neutrophil extracellular traps (NETs) were added (now lines 40-42).

3. In materials and methods, providing a brief description of the process of neutrophil isolation would enhance the reader's understanding and provide the necessary context.

As requested, details regarding the steps used to isolate neutrophils from peripheral blood were added to the Methods (lines 75-80).

4. Please include the statistical tests used in all figure legends (Fig.1, 2, 3, 4, 5 and S4).

Tests used for statistical analysis are now provided in each figure legend. This is also specifically stated in the Methods (lines 279-280).

5. Fig 2 B, C, D, E, F, G: Adding the word days to the x-axis will improve the comprehension of the figure. Although the results are strong, the order of the graph in this figure is confusing; please reorganize it for better visualization. In the legend, it would also be helpful to have the word 1-5 days in QVD-treated neutrophils (Q1-Q5). We agree that the layout of Figure 2 was confusing. The figure panels were rearranged for clarity with blot quantitation graphs now immediately adjacent to the blots themselves. Wording was added to the legend to indicate that X -axes refer to freshly isolated (P0), 1-2 day aged neutrophils (P1, P2) or cells treated with QVD for 1-5 days (Q1-Q5). Finally, we changed the Y-axis label in panel E from ‘luminescence’ to ‘Caspase Activity (RLU)’ to increase clarity. RLU was defined as ‘relative light units’ in the figure legend.

6. Fig 3. The order graph is confusing; please reorganize it for better visualization.

Thank you for this suggestion. The layout of the figure panels was rearranged for clarity. In addition, the representative JC-1 flow plots were moved to Supporting information (S4 Fig). To address comments of Reviewer 2, new data were added (panels 3F (MitoSox staining), 3G (GSh/GSSG ratios), and 3H (total NAD).

7. Please please add the size bar information in all micrographs.

Size bars were added to all microscopy images.

8. Supplementary Fig. 1: Please specify the abbreviation used in the graph: (Q6-Q10).

The abbreviation Q6-Q10 was defined as 6-10 days of QVD treatment in the legend.

9. In the results, please state and further explain the changes in instantaneous velocity and chemotactic index. It would also be beneficial to discuss the implications of these findings specifically for the bigger picture.

We appreciate this suggestion and apologize for the lack of detail. We added additional detail regarding the assay and how the movies of cell migration were analyzed to calculate speed and directionality was added to the Methods (lines 241-248). Additionally, our interpretation of the data was revised and expanded as indicated in the Results (lines 464-466) and the Discussion (lines 672-676)

 with regard to the bigger picture.

Reviewer #2:

Synopsis:

The manuscript investigates the effects of the pan-caspase inhibitor Q-VD-OPh on human neutrophil lifespan and functionality. The authors show that Q-VD-OPh significantly extends neutrophil viability by preventing apoptosis for five days while maintaining functional capacities such as phagocytosis and chemotaxis. However, mitochondrial depolarization still occurs despite the inhibition of apoptosis.

Strengths:

Comprehensive apoptosis inhibition analysis: The study uses multiple assays to confirm the inhibition of apoptosis, providing robust evidence.

Functional focus: The paper evaluates not only apoptosis inhibition but also crucial neutrophil functions, ensuring a more comprehensive understanding of cell survival.

Clinical relevance: Extending neutrophil lifespan has potential therapeutic implications in conditions where neutrophil function is compromised.

Weaknesses:

Incomplete mechanistic insight: While the inhibition of caspases is well demonstrated, the underlying molecular mechanisms explaining the preservation of function despite mitochondrial dysfunction need further elaboration.

Lack of in vivo validation: All experiments are performed in vitro, limiting the translatability of the findings to real-life physiological scenarios.

Neutrophil viability versus functionality: The study highlights that while apoptosis is inhibited, mitochondrial depolarization still occurs, raising questions about the long-term viability and effectiveness of these cells in a physiological context.

Critical comments for Major Revision:

1. Impact of inhibitor on Mitochondrial Function: 

To evaluate the effect of Q-VD-OPh on mitochondrial function, focus on mitochondrial membrane potential, ROS production, and redox balance. Use JC-1 staining to assess membrane potential and analyze changes in mitochondrial depolarization. For ROS, employ both flow cytometry (DHE or MitoSOX) and luminol-based assays to detect superoxide production. Additionally, measure mitochondrial redox state by monitoring NADH/NAD+ ratios or GSH/GSSG ratios, which will indicate oxidative stress levels and mitochondrial health.

We appreciate these constructive comments and performed additional experiments to define in greater detail the effects of Q-VD-OPh on processes related to mitochondrial function. Analysis of mitochondrial depolarization by JC-1 staining is shown in Figure 3C and representative flow cytometry plots are now shown in S4 Fig. As suggested, we used MItoSox to quantify mitochondrial superoxide using flow cytometry. These data are shown in new Figure 3F. Data for NADPH oxidase derived ROS, detected using the luminol chemiluminescence assay are shown in Figure 6. With respect to redox state, we confirmed that total NAD is very low in neutrophils (new Fig 3H) and as suggested also quantified GSH/GSSG ratios (new Fig 3G). Please see updated Methods, Results and Discussion (lines 186-187, 191-205, 437-450, 587-590).

2. Inhibitor’s Role in Cellular Pathways Like Autophagy: 

To explore whether Q-VD-OPh affects autophagy, perform western blotting to detect autophagy markers such as LC3-II, p62, and Beclin-1. Combine this with transmission electron microscopy (TEM) to visualize autophagosomes. Use inhibitors like chloroquine or bafilomycin A1 to evaluate autophagic flux in combination with Q-VD-OPh. Additionally, assess the activity of mTOR (key autophagy regulator) via its phosphorylation status using phospho-specific antibodies. This will clarify whether Q-VD-OPh modulates autophagy to support neutrophil survival.

Again, we appreciate these comments and perfomed additional experiments to better define the influence of Q-VD-OPh on autophagy in neutrophils (new S6 Fig). We quantified phospho- and total mTOR in single cells using flow cytometry. These data show a trend to lower total and phosphor-mTOR in 48h aged neutrophils, but the ratios for all conditions were similar and did not significantly differ. We used confocal microscopy to detect p62+ autophagosomes +/- Bafilomycin treatment and immunoblotting to detect LC3B-I andLC3B-II. These data are shown in new S6C Fig and S6B Fig. Overall, the data suggest that autophagy is low under the conditions of this study and is not significantly altered in aged cells or following QVD treatment, and as such is not a major determinant of PMN survival during days 1-5 of QVD treatment. However, this does not preclude a role for autophagy on QVD day 6-10. The Methods have been updated, including lines 130-137, Results lines 558-581 and Discussion lines 618-627. We were unable to include TEM images as the University of Missouri Microscopy Core suffered catastrophic flooding and is currently closed for repairs and is awaiting replacement instrumentation installation. 

3. Effects on Neutrophil Behavior: 

Investigate if Q-VD-OPh alters the natural behavior of neutrophils by studying their chemotaxis, phagocytosis, and degranulation. Use EZ-TAXIScan assays to evaluate neutrophil migration towards fMLF gradients. For phagocytosis, opsonized zymosan assays combined with fluorescence microscopy will quantify particle uptake. Assess degranulation by measuring extracellular levels of enzymes like MPO or elastase using ELISA. These functional assays will reveal if Q-VD-OPh preserves normal neutrophil behavior or induces functional impairments.

All these neutrophil functions were assayed. Chemotaxis of cells in fMLF gradients was analyzed using EZ-TAXIScan video imaging and these data are shown in Figure 4. Phagocytosis of Opsonized zymosan was quantified by light microscopy and is shown in Figure 5. As suggested, we performed additional experiment to quantify elastase release by ELISA as an indicator of degranulation (new Figure 7). See also updated text on lines 261-269, 519-523 and 680 and the Abstract line 18.

---

## [Decision Letter · Decision Letter 1]

19 Dec 2024

Effects of the pan-caspase inhibitor Q-VD-OPh on human neutrophil lifespan and function

PONE-D-24-41135R1

Dear Dr. Allen,

We’re pleased to inform you that your manuscript has been judged scientifically suitable for publication and will be formally accepted for publication once it meets all outstanding technical requirements.

Kind regards,

Saeid Ghavami, PhD

Academic Editor

PLOS ONE

Additional Editor Comments (optional):

Reviewers' comments:

Reviewer's Responses to Questions

**Comments to the Author**

1. If the authors have adequately addressed your comments raised in a previous round of review and you feel that this manuscript is now acceptable for publication, you may indicate that here to bypass the “Comments to the Author” section, enter your conflict of interest statement in the “Confidential to Editor” section, and submit your "Accept" recommendation.

Reviewer #1: All comments have been addressed

Reviewer #2: All comments have been addressed

2. Is the manuscript technically sound, and do the data support the conclusions?

Reviewer #1: Yes

Reviewer #2: Yes

3. Has the statistical analysis been performed appropriately and rigorously? 

Reviewer #1: Yes

Reviewer #2: Yes

4. Have the authors made all data underlying the findings in their manuscript fully available?

Reviewer #1: Yes

Reviewer #2: Yes

5. Is the manuscript presented in an intelligible fashion and written in standard English?

Reviewer #1: Yes

Reviewer #2: Yes

6. Review Comments to the Author

Reviewer #1: (No Response)

Reviewer #2: I really enjoyed the careful response and revision by the respected authors. All comments are correctly addressed. I do not have any further suggestion. Including mechanistic data and functional data of the inhibitor makes this research paper very important contribution to the field.

7. PLOS authors have the option to publish the peer review history of their article (what does this mean?). If published, this will include your full peer review and any attached files.

Reviewer #1: No

Reviewer #2: No

---

## [Editor Report · Acceptance letter]

23 Dec 2024

PONE-D-24-41135R1 

PLOS ONE

Dear Dr. Allen, 

I'm pleased to inform you that your manuscript has been deemed suitable for publication in PLOS ONE. Congratulations! Your manuscript is now being handed over to our production team.

Kind regards, 

on behalf of

Dr. Saeid Ghavami 

Academic Editor

PLOS ONE